# Transepithelial Enhanced Fluence Pulsed Light M Accelerated Crosslinking for Early Progressive Keratoconus with Chemically Enhanced Riboflavin Solutions and Air Room Oxygen

**DOI:** 10.3390/jcm11175039

**Published:** 2022-08-27

**Authors:** Cosimo Mazzotta, Ashraf Armia Balamoun, Ayoub Chabib, Miguel Rechichi, Francesco D’Oria, Farhad Hafezi, Simone Alex Bagaglia, Marco Ferrise

**Affiliations:** 1Departmental Ophthalmology Unit, Alta Val d’ Elsa Hospital, USL Toscana Sudest, Post Graduate Ophthalmology School, University of Siena, Siena Crosslinking Center Siena Italy, 53100 Siena, Italy; 2Watany Research and Development Center, Ashraf Armia Eye Clinic and Al Watany Eye Hospital, Cairo 11511, Egypt; 3Department of Ophthalmology, Cornea and Refractive Surgery, San Rossore Medical Center, 56122 Pisa, Italy; 4Centro Polispecialistico Mediterraneo, 88050 Sellia Marina, Italy; 5Ophthalmology Unit, Santa Maria Hospital-GVM, 70125 Bari, Italy; 6ELZA InsZurich, 8953 Dietikon, Switzerland; 7Laboratory for Ocular Cell Biology, University of Zurich, 8006 Zurich, Switzerland; 8Roski Eye Institute, University of Southern California, Los Angeles, CA 90001, USA; 9Department of Ophthalmology, University of Wenzhou, Wenzhou 325015, China; 10Ophthalmology Unit, USL Toscana Sud-Est, 52100 Massa Marittima, Italy; 11Studio Oculistico Ferrise, 88046 Lamezia Terme, Italy

**Keywords:** keratoconus, crosslinking, accelerated crosslinking, riboflavin, transepithelial, enhanced fluence corneal cross linking, collagen crosslinking, Epi-On crosslinking, corneal crosslinking, ectasia

## Abstract

Purpose: To assess the 3-year clinical results of the 18 mW 7 J/cm^2^ transepithelial enhanced fluence pulsed light M accelerated crosslinking in the treatment of progressive keratoconus (KC) with chemically enhanced hyper-concentrated riboflavin solutions without iontophoresis and with air-room oxygenation. Setting: Siena Crosslinking Center, Siena, Italy. Methods: Prospective pilot, open non-randomized interventional study including 40 eyes of 30 young adult patients over 21 years old (10 simultaneous bilateral) with early (Stage I and II) progressive KC undergoing TE-EFPL 18 mW/7 J/cm^2^ ACXL (EFPL M TECXL). The 12 min and 58 s pulsed light (1 s on/1 s off) UV-A exposure treatments were performed with a biphasic corneal soaking using Paracel I 0.25% for 4 min and Paracel II 0.22% for 6 min riboflavin solutions and New KXL I UV-A emitter (Glaukos-Avedro, Waltham, USA) at an air room of 21% oxygenation. All patients completed the 3-year follow-up. Results: CDVA showed a statistically significant improvement in the third postoperative month (Δ + 0.17 d. e.) with a final gain of +0.22 d. eq. AK showed a statistically significant decrease in the sixth postoperative month (Δ − 1.15 diopters). K itmax showed a statistically significant decrease at 1-year follow-up (Δ − 1.3 diopters). The coma value improved significantly by the sixth month (Δ − 0.54 µm). MCT remained stable during the entire follow-up. No adverse events were recorded. Corneal OCT revealed a mean demarcation line depth at 282.6 ± 23.6 μm. Conclusions: Transepithelial enhanced fluence pulsed light M accelerated crosslinking with chemically enhanced riboflavin solution halted KC progression in young adult patients without iontophoresis and no intraoperative oxygen supplementation addressing the importance of increased fluence.

## 1. Introduction

Riboflavin/ultraviolet-A (UVA) corneal collagen cross-linking (CXL) with epithelium removal (Epi-Off CXL) represents the gold standard of conservative treatment of progressive keratoconus (KC) and iatrogenic corneal ectasias [1,2].⁠ Epithelium removal represents a fundamental step enabling riboflavin and oxygen diffusion into the corneal stroma, which are necessary to optimize the photo-oxidative reaction, leading to an efficient photodynamic accelerated corneal crosslinking (ACXL) [1,3,4].⁠ As the removal of the epithelial barrier exposes the cornea to some risks such as infectious keratitis, wound healing stimulation (haze and thinning), and temporary glare disability, making the visual recovery longer and often associated to greater patient’s discomfort [5,6,7,8]⁠, the research is focusing on “enhanced” or “compensated” transepithelial (Epi-on) ACXL protocols with higher energy (fluence) over the standard 5.4 J/cm^2^ set in the Dresden protocol (e.g., the enhanced fluence pulsed light Epi-on iontophoresis protocols recently developed by Mazzotta et al). Despite a barrier effect of the corneal epithelium lowering riboflavin and oxygen diffusion [5,9,10,11,12,13], several strategies have been recommended to increase epithelial permeability to riboflavin such as the mechanical focal disruption of the epithelium, preoperative administration of preservative-based anesthetic drops, and the application of 20% alcohol solution. Laboratory tests showed an inhomogeneous stromal riboflavin loading comprised between 20% and 50% compared with the gradient of concentration of riboflavin achieved into the stroma after epithelium removal [5,9,13]. Of course, this single parameter does not imply an undoubted reason of ineffectiveness [14] of the treatment since crosslinking physiochemical pathways are complex and given by the combination of multiple interactions including the fluence, UV-A ray absorption, mode of UV-light exposure, oxygen availability and diffusion, the UV-A-riboflavin interaction with the collagen and proteoglycan substrate as well as the riboflavin gradient of concentration. Clinical, biologic, and genetic factors such as allergy, eye-rubbing, and hormones influence the specific response to treatment on a case-by-case basis [15,16]. Attempts have been made to improve the penetration of riboflavin through the epithelium through the addition of chemical agents and additives (enhancers) such as benzalkonium chloride (BAC), trometamol (TRIS), and ethylenediamine tetra acetic acid (EDTA) [10,13,15]. Despite these enhancers, clinical studies using transepithelial riboflavin formulations have shown contradictory results, most of which were inferior compared to Epi-off CXL in stabilizing the corneal ectasia in the long-term follow-up, especially in younger adult patients [9,10,11]. Transepithelial riboflavin loading was further improved by iontophoresis, being riboflavin water-soluble and negatively charged at physiological pH, by the application of a low intensity electrical gradient of 1 m ampere × 5 min. Electrically assisted iontophoresis forced the permeation of the riboflavin into the stroma at a higher level compared with the original Epi-On procedures with chemically enhanced solutions, allowing an average riboflavin concentration two-fold higher throughout the whole corneal depth, with deeper and more homogeneous stromal distribution, even if its concentration was found halved compared with passive diffusion after Epi-off CXL [9,13,14]. Despite the increased intra-stromal concentration of riboflavin, the long-term follow-up of the original 5.4 J/cm^2^ iontophoresis CXL (I-CXL) showed a 26% failure rate [17]. Mazzotta et al. improved the results of the I-CXL method by increasing the fluence to 7 J/cm^2^, pulsing the light illumination and washing the riboflavin biofilm away from the corneal surface before starting the UV-A light, thus eliminating the riboflavin shielding effect beyond the epithelium itself, thus improving the efficacy of the treatment and the depth of the demarcation line. This novel approach is known as enhanced fluence pulsed light iontophoresis or the new iontophoresis protocol [18,19]. Another important step forward in increasing the efficacy of Epi-on treatment by conferring it a refractive empowerment, beyond increased fluence and pulsed light, was obtained by using the intraoperative oxygen supplementation as documented for the first time by Mazzotta et al. [20]. The aim of this study was to validate the efficacy of the enhanced fluence (7 J/cm^2^) pulsed light (1 s on, 1 s off) M Epi-on crosslinking protocol (EFPL-M-TECXL) in a series of young adult patients affected by progressive KC with chemically enhanced riboflavin solutions and without intraoperative oxygen supplementation in order to simplify the procedure, reducing the minor invasiveness of iontophoresis loading and the adjunctive costs of the iontophoresis and oxygen kit, while maintaining an overall efficacy of the method in stabilizing keratoconus in the mid- to long-term follow-up of 3 years.

## 2. Methods

### 2.1. Surgical Procedure

The EFPL-M-TECXL technique is displayed in Table 1. The treatment was performed under topical anesthesia with 4% oxibuprocaine chlorydrate 1.6 mg/0.4 mL drops, applied 5 min before the treatment (one drop each minute) and after applying a closed valve eyelid speculum, which consisted in epithelial surface swiping with a sponge soaked with 0.25% Paracel 1 solution followed by a biphasic (4 min part one and 6 min part two) of riboflavin soaking with Paracel 1 (Glaukos-Avedro, Waltam, MA, USA) 0.25% riboflavin solution, dropped each minute for 6 min and Paracel 2 (Glaukos-Avedro, Waltam, MA, USA) 0.22% dropped each 30 s for 4 min. Abundant irrigation of the corneal surface was performed for 15–20 s with a balanced salt solution (BSS) before starting the UV-A irradiation with the New KXL I UV-A emitter (Glaukos-Avedro, Waltam, MA, USA), adopting a higher fluence of 7 J/cm^2^ for a pulsed light mode of exposure (1 s on 1 s off cycle) and a total exposure time of 12 min and 58 s at 18 mW/cm^2^ UV-A power, in conformity with the EFPL-iontophoresis technique, without adding intraoperative supplementary oxygen and avoiding the invasiveness of iontophoresis imbibition [18,19]. At the end of the procedure, the cornea was medicated with two drops of preservative-free netilmicin plus dexamethasone, cyclopentolate, and ketorolac eye drops plus Carbopol 974 P 0.25% gel, dressing the eye with a therapeutic soft contact lens bandage for 24 h to protect the epithelium. After therapeutic contact lens removal (the day after), fluormetholone 0.2% eye-drops tapered four times/day in the first week, three times in the second week, and two times in the third week. Sodium hyaluronate 0.2% lacrimal substitutes were administered for 6 to 8 weeks. Oral NSAID (10 mgr Keratorolac) was prescribed in the first 24 h in the case of pain in combination with Ketorolac eye drops administered four times × day.

### 2.2. Dataset, Study Design, and Inclusion Criteria

The prospective open non-randomized interventional study was unanimously approved by the Institutional Review Board (IRB) of the Siena Crosslinking Center, Siena, Italy (EFPL M TECXL Protocol 2.0), in accordance with the ethical principles set in the Declaration of Helsinki, adopting 0.25% and 0.22% enhanced riboflavin solutions just approved and available on the market for transepithelial (Epi-ON) crosslinking use (Paracel I and II, Glaukos-Avedro, Waltam, MA, USA) and irradiation parameters only used in the EFPL-I CXL (New Iontophoresis) protocol [19]⁠.

The study included 40 eyes of 30 young adult patients over 21 years of age (10 simultaneous bilateral treatments), of which 34 were males (85%) and six females; every eye was affected by early progressive Stage I and II keratoconus (KC) according to the Amsler-Krumeich KC staging [21]⁠. In 25 eyes (62.5%), the treatment involved the less affected eye after having performed an epithelium-off (Epi-off) accelerated 9 mW/5.4 J/cm^2^ crosslinking in the worst eye [4,22], while in 15 eyes (37.5%), the treatment was performed in the worst eye as the first choice. The mean age of the study cohort was 28.2 ± 4.9 years. The progression of KC was defined as an increase in the K max ≥ 1 diopter (D) using the Scheimpflug–Placido corneal tomography system Sirius, Costruzione Strumenti Oftalmici (C.S.O.), Florence, Italy; minimum corneal thickness (MCT) reduction ≥10 µm; worsening of uncorrected distance visual acuity (UDVA) and corrected distance visual acuity (CDVA) ≥0.1 decimal equivalent or change of ≥0.5 in mean refractive spherical equivalent (MRSE) in the last 6 months of clinical and instrumental observation [21]. MCT was at least 400 µm (epithelium included). Clear corneas with no sub-apical opacities or scars, no history of previous HSV and other infectious keratitis or autoimmune diseases, and no severe dry eye were included. All patients signed a specific written informed consent. The demographic data are displayed in Table 2.

### 2.3. Measurements and Devices

Ophthalmic evaluations were performed before CXL and at all follow-up visits (1, 3, 6, 12, 24, and 36 months). The evaluation included the uncorrected distance visual acuity (UDVA), best spectacle corrected distance visual acuity (CDVA), and the biomicroscopic corneal examination, OSDI test, and noninvasive topographic break-up time test (NI-BUT) for excluding dry-eye disease. Scheimpflug based corneal tomography (Sirius, CSO, Florence, Italy) was used to measure the maximum curvature simulated K reading (Kmax), high-order aberration (Coma), minimum corneal thickness (MCT), and apical curvature (AK). Anterior segment optical coherence tomography (AS-OCT) with the I-Vue (Optovue, Freemont, CA, USA) was performed to assess the demarcation line depth at the first post-operative month.

### 2.4. Statistical Analysis

According to the study purpose, the follow-up examination was performed at 24 h, 1, 3, 6, 12, 24, and 36 months. All patients completed the 3 year follow-up. A two-tailed paired samples t-test was used to compare each baseline measurement with the respective follow-up measurements. Differences with *p* < 0.05 were considered as statistically significant. Data were collected and analyzed with PRISM 6.0 GraphPad Software (La Jolla, CA, USA).

## 3. Results

The uncorrected distance visual acuity (UDVA) did not show statistically significant changes, while the spectacles corrected distance visual acuity (CDVA) showed an improvement, becoming statistically significant at the third postoperative month (Δ + 0.2 ± 0.08 decimal equivalents), *p* < 0.05 (Figure 1).

The apical curvature (AK) flattened significantly, becoming statistically significant at 1 year follow-up (Δ − 1.59 D ± 0.8), *p* < 0.05, showing stability during the follow-up (Figure 2).

The maximum simulated keratometry value (K Max) improved along the follow up since the third month, becoming statistically significant at 1 year (Δ of −1.3 D ± 0.78 D) from the baseline value (Figure 3).

The coma value showed a statistically significant improvement at the sixth month (Δ − 0.54 µm ± 0.2), continuing thereafter (Figure 4).

The postoperative spectral-domain corneal OCT performed one month after treatments revealed a clear demarcation line with a mean depth of 278 ± 32 μm (Figure 5).

No adverse events such as haze or infections were recorded during the follow-up, except for two cases of paracentral apical de-epithelialization requiring aa therapeutic soft contact lens bandage for 72 h. A punctate epitheliopathy was found in all cases at fluorescein dye test resolving after 24 to 48 h with a soft contact lens bandage and lubricants. According to the Visual Analogue Scale (VAS) pain scale, the average value as reported in our series was 5 ± 2 in the first 4–6 postoperative hours was 5 (from mild to moderate pain), rapidly dropping to 0 (no pain) after the fifth to the sixth postoperative hour.

## 4. Discussion

The 3-year results of the EFPL M CXL meet the goal of improving the efficiency and feasibility of the Epi-On CXL. The fact that trans-epithelial protocols are considered worldwide the “*way to go of CXL*” is largely based on their benefits in eliminating the risk of infectious keratitis, the stimulation of stromal scarring, and consequent extreme thinning, speeding up the functional patient recovery, reducing the postoperative pain duration, and minimizing the microstructural damage to the ocular surface, thus quickly rehabilitating the patient to study and work activities in a few days [6,7,8]. Moreover, this treatment allows for the possibility of carrying out the procedure comfortably in outpatient mode while reducing the invasiveness and adjunctive kit costs, see Table 3. Nevertheless, in consideration of the high safety profile in the absence of ascertained clinical worsening, this protocol paves the way to the preventive use of crosslinking and to bilateral simultaneous treatments, also in combination with Epi-off CXL performed in the worst eye.

The novelty of this EFPL M TECXL protocol is the use of chemically enhanced 0.25% and 0.22 % riboflavin solutions instead of iontophoresis imbibition and without intraoperative oxygen supplementation. This simplified Epi-on approach, avoiding iontophoresis and supplementary oxygen kits resulted in being less invasive and more cost-effective, also offering the possibility of a full outpatient treatment, thus optimizing the clinic workflow. What is known is that the epithelium shield for short UV-A radiations at the 370 nm waveband is 30%, so by increasing the fluence at 7 J (30% over the standard 5.4 J set in the Dresden protocol), we compensated the epithelial UV-A photo-absorption. Moreover, we also know that epithelium oxygen consumption is around 40%, so including the pulsed light while increasing the fluence and the concentration of the riboflavin solution at 0.25% and 0.22%, respectively, we enhanced the aerobic and anaerobic kinetic of CXL, increasing its overall physiochemical impact. The results of this study demonstrate that by performing the treatment using 18 mW/cm^2^ UV-A power for 12 min and 58 s of UV-A pulsed light exposure, we halted KC progression without adverse events, similarly to the EFPL I-CXL protocol in a 3-year follow-up, of which the present requirements are a simplified evolution that works at air room oxygen conditions [18,19]. On the other hand, we have recently demonstrated that by using 30 mW/cm^2^ UV power with a shorter exposure time, the intraoperative oxygen supplementation enhances the aerobic pathway of CXL, increasing the treatment penetration and corneal flattening [20]⁠. In this case, the 18 mW/cm^2^ UV-A power and the longer exposure time optimizes the CXL kinetic, avoiding the necessity of supplementary intraoperative oxygen that is recommended with 30 mW trans-epithelial customized ACXL protocols [20]; in fact, the protocols being equal, the use of supplementary oxygen was found to also give the treatment some refractive power by increasing the flattening of the treated cornea [22,23,24,25].

A limitation of the technique could be the reduced amount and inhomogeneous stromal distribution of riboflavin and the epithelial photo-attenuation of the UV photons impact, which should require a further fluence increase, however, the demarcation line [26] detection and the clinical results after a 3 year follow-up showing keratoconus stability and functional results similar of those obtained with EFPL I-CXL seem to minimize this issue [19].⁠ This may be due to the complex kinetic and different pathways of CXL as reported by Lin et al., showing that CXL includes different aerobic and anaerobic pathways so it can also occur in low oxygen environment (Type-I) as well as in an oxygen dependent Type-II pathway [27].⁠ Multiple mixed reactions co-exist in the CXL dynamic, allowing for the enhancement of the anaerobic kinetic by acting directly on the collagen-proteoglycans substrate by increasing the riboflavin concentration to 0.25% and the fluence to 7 Joule/cm^2^, thus increasing the whole kinetic of the Epi-On accelerated methods [19,20,22]. Of course, there is no possible Type-II CXL when oxygen is depleted or in a non-oxygen environment [28]. Indeed, the Type-II CXL is strictly oxygen mediated and its efficacy is proportional to the light dose and oxygen and intra-stromal riboflavin concentration, while the Type-I CXL effect is mainly related to the treatment energy dose and riboflavin concentration gradient, which must be higher and well-distributed in about 200 µm of the corneal stroma, allowing for direct coupling with the collagen substrate [27]. These mixed physio-chemical reactions may explain the efficacy of the new transepithelial treatments with increased fluence, higher riboflavin concentration solutions, and pulsed-light [19,23,25]. Higher fluence (10 J/cm^2^) and chemically boosted or enhanced riboflavin solutions further increase the efficiency of trans-epithelial CXL in a low oxygen environment and anaerobic conditions. Indeed, the efficacy of CXL is proportionally dependent from fluence, as we have just reported in our pivotal epithelium-off topography-guided ACXL treatments [29] and in customized epithelium-on treatments [20].

Increasing the fluence increases the depth of the demarcation line, substantially increasing the treatment volume and the biomechanical power of crosslinking itself [20,26,29].

It was demonstrated [30,31,32] that the influence of CXL is characterized by a mechanism-based strain energy function, which not only explicitly depends on the density of the cross-links (as a function of the corneal thickness and UVA irradiation dose), but also relies on the discrepant distribution of cross-links in the proteoglycan matrix and along the fibers. The inflation test experimental data for the IOP-apex displacement relationships of porcine cornea at different irradiation doses of 2.7 J/cm^2^, 8.10 J/cm^2^ and 10.8 J/cm^2^, respectively, showed good collimation between the predicted and simulated results, thus addressing that at the same value of IOP, the apex displacement of corneal materials after cross-linking is obviously much smaller than that without cross-linking. The simulated results are able to predict the macroscopic IOP apex displacement relations as a function of the UVA irradiation dose. Therefore, the cornea after CXL can better resist the geometrical deformation when stimulated by the increase in IOP and the apex displacement further becomes smaller with the increasing irradiation dose, and is determined by the inhomogeneous increment of cross-link density from the anterior layer of the corneal stroma. For the depth-dependent density of the cross-links, the increment rate of the cross-link density with the irradiation dose was the fastest in the anterior cornea stroma and the highest at 10.8 J/cm^2^. Therefore, the gradient change in the material strength became increasingly obvious for the cross-linked corneas with a comparatively high value of irradiation dose. The finite element analysis provided at increasing irradiation doses proved that the increase in the cross-link density at a comparatively high value of irradiation dose could help improve the material strength, thus increasing the fluence not only provides an increased depth in the demarcation lines, but an increased cornea resistance that is crucial in Epi-On treatments. These clinical and experimental observations confirmed that the way to go for Epi-On CXL treatments is to find a balance between the riboflavin concentration and higher fluence between 8.1 and 10.8 J/cm^2^ [30,31,32,33].

Our preliminary functional and morphological data in a new Epi-On study protocol with chemically enhanced riboflavin solutions and 10 J/cm^2^ fluence in progress at the Siena CXL Center, Italy, are showing promising evidence of better results.

As of now, the patient categories suggest that in the clinical practice of this novel protocol of treatment for corneal ectasia, young adults over 26 year of age are treated in both eyes, even conducted on the same day to reduce the burden of time and organization for both the patient and surgeon, and young adults over the age of 21 but only in the eye with less advanced keratoectasia.

This treatment protocol can possibly be repeated if the progression of the ectasia restarts during the follow-up, still keeping the advantages of Epi-On procedures, that is to say, virtually absent risks of infection and uncontrolled corneal scarring.

In patients affected by progressive keratoconus who are less than 21 years of age, the gold standard of treatment still remains an accelerated continuous light 9 mW 5.4 J/cm^2^ Epi-OFF CXL [4].

Further studies, larger patient cohorts, and parameter implementations are necessary to improve the efficacy of epithelium preserving CXL protocols, making it highly effective, more accessible, and feasible in outpatient modality.

Moreover, enhanced Epi-On CXL paves the way to a total prophylactic use of CXL, allowing us to become active providers instead of awaiting the loss of corneal resistance and visual decay, also in forme fruste keratoconus (FFKC) and suspicious ectasia cases, changing our current mindset.

## Figures and Tables

**Figure 1 jcm-11-05039-f001:**
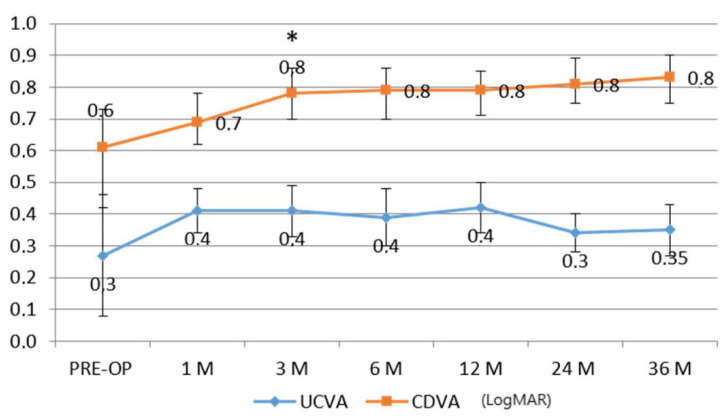
The uncorrected distance visual acuity (UDVA) did not show statistically significant changes, while the spectacles corrected distance visual acuity (CDVA) showed an improvement, becoming statistically significant at the third postoperative month (Δ + 0.2 ± 0.08 decimal equivalents).

**Figure 2 jcm-11-05039-f002:**
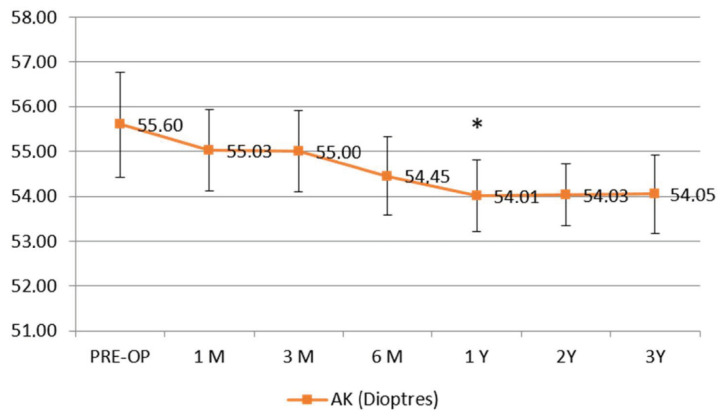
Apical curvature (AK). Topographic apical curvature (AK) showed a statistically significant decrease at 1 year follow-up (Δ − 1.59 ± 08 diopters), reaching a stabilization.

**Figure 3 jcm-11-05039-f003:**
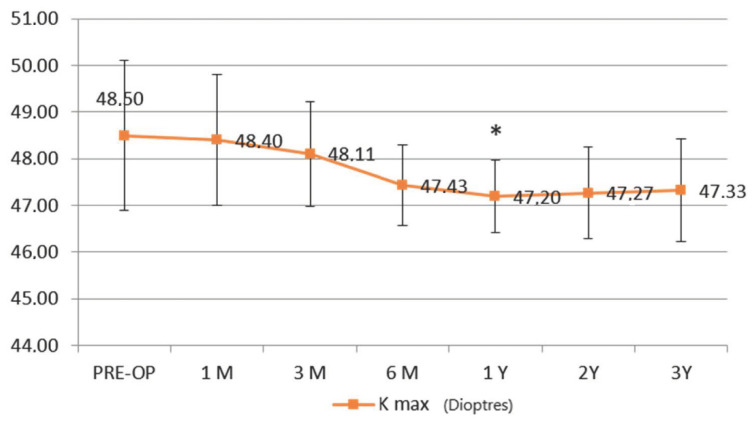
Maximum keratometry (K max). Topographic simulated maximum K reading (K Max) showed a progressive decrease starting at the third postoperative month and becoming statistically significant at one year follow-up (Δ − 1.3 ± 0.78 D diopters), showing a stability during the entire follow-up.

**Figure 4 jcm-11-05039-f004:**
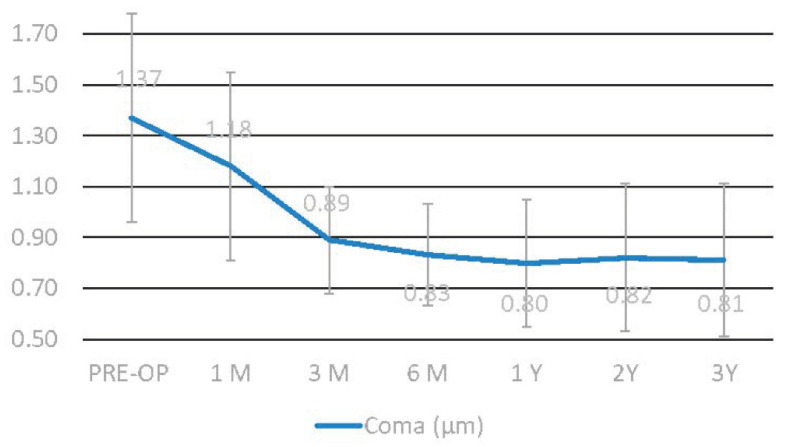
The coma value showed a statistically significant improvement at the sixth month (Δ − 0.54 µm ± 0.2), continuing thereafter.

**Figure 5 jcm-11-05039-f005:**
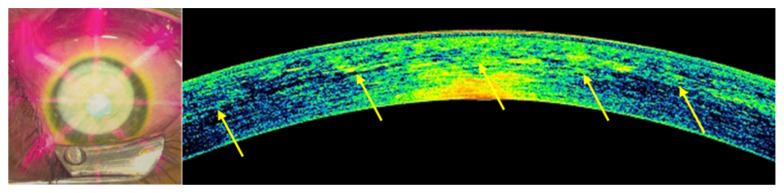
The postoperative spectral-domain corneal OCT performed one month after the treatments revealed a clear demarcation line (yellow arrows) with a mean depth of 278 ± 32 μm.

**Table 1 jcm-11-05039-t001:** Enhanced fluence pulsed light M Epi-on crosslinking protocol (EFPL-M-TECXL).

Parameter	Variable
Treatment target	KC stabilization
Fluence (total) (Joule/cm^2^)	7 Joule/cm^2^
Soak time and interval (minutes)	Paracel I (Part one 4 min) + Paracel II (Part two 6 min)
Intensity (mW)	18 mW/cm^2^
Irradiation Time	12 min and 58 s
Epithelium status	On
Chromophore	Riboflavin
Chromophore carriers	Trometamol, Na-EDTA, no Dextran
Chromophore osmolarity	Isotonic + hypotonic
Chromophore concentration	0.25% (part one) + 0.22% (part two)
Light source	New KXL I (Glaukos-Avedro, Waltam, MA, USA)
Irradiation mode (interval)	Pulsed (1 s on–1 s off)
Protocol modifications	EFPL I-CXL
Protocol abbreviation	EFPL M-TECXL

**Table 2 jcm-11-05039-t002:** The baseline demographic data.

Baseline Characteristics 40 Eyes of 30 Pat	Value (Mean)	SD or %
**Mean Age (Years)**	28.2	±4.9
**Male**	34	85%
**UDVA d. eq.**	0.27	±0.12
**CDVA d. eq.**	0.61	±0.19
**Kmax (D)**	48.52	±1.63
**Coma (D)**	1.37	±0.41
**AK (D)**	55.61	±1.17
**Minimum corneal** **Thickness µm**	467.43	±17.27

**Table 3 jcm-11-05039-t003:** The advantages of enhanced trans-epithelial CXL protocols.

Reduction/elimination of corneal infection risk
Reduction/elimination of corneal wound healing stimuli (haze, scarring, extreme thinning)
Faster patient recovery and visual rehabilitation
Minimization of microstructural damage to the ocular surface
Prevention of dry eye preserving the nerve plexus structure
Simultaneous bilateral treatment
Quick rehabilitation of the patient to school and work activities
Full outpatient procedure
Reduced costs
**Indications** **of Enhanced Trans-Epithelial CXL Protocols**
Preventive use of CXL without awaiting progression
Forme fruste keratoconus (FFKC)
Suspicious ectasia

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
