# Peer review of "Transepithelial Enhanced Fluence Pulsed Light M Accelerated Crosslinking for Early Progressive Keratoconus with Chemically Enhanced Riboflavin Solutions and Air Room Oxygen"

_jcm, 2022, doi:10.3390/jcm11175039_

Round 1

Reviewer 1 Report

* the figures could be easier to read. Numbers overlap, and one is too dark to read at all. 

* figure 5, what is the explanation for the bright zone in the posterior central cornea? 

* 209-211, are you saying patients felt zero pain only a few hours after the procedure? Also, is there any information on epi damage after the procedure? Rinsing the riboflavin layer would not mitigate all damage. 

*230-233, do you have any references or reasoning for this? Explain further. 

* paragraph beginning at 247, reword this, it is very confusing. The first sentence especially so. 

* multiple grammar and typo errors 

Author Response

Thank you for taking your precious time to help us with this manuscript.

About your questions:

1 *the figures could be easier to read. Numbers overlap, and one is too dark to read at all. 

Thanks for the suggestion. We will try to make it more readable in the final draft of the manuscript.

2 * figure 5, what is the explanation for the bright zone in the posterior central cornea? 

Thanks for the question. The highly reflective zone in the posterior part of the cornea is just an artifact of the OCT used. It can be seen also in untreated corneas, not just post corneal CXL. To better help the reader discern the demarcation line, yellow arrows were used to mark it.

3 * 209-211, are you saying patients felt zero pain only a few hours after the procedure? Also, is there any information on epi damage after the procedure? Rinsing the riboflavin layer would not mitigate all damage. 

The patients reported moderate pain in the postoperative period. The pain was on average very mild to absent after 6 hours, with the aid of a bandage contact lens and a generous use of gel lubricants.

About the epithelium, it is a very interesting topic and indeed we plan to write a paper about the condition of the epithelium (we are already gathering data) after this procedure in the near future.

4 *230-233, do you have any references or reasoning for this? Explain further. 

This simplified Epi-On approach, avoiding iontophoresis and supplemental oxygen kits resulted less invasive and more cost-effective, also offering the possibility of a full outpatient treatment, thus optimizing the clinic workflow.”

Thanks for the question, this method of corneal CXL does not need neither the “iontophores kit” nor the “supplemental oxygen kit”. This means that the procedure is less costly (no need to buy ulterior equipment for every procedure) and simpler for the surgeon and the staff (no need to order special kits, no need to create a suction on the cornea in case of iontophoresis).

5 * paragraph beginning at 247, reword this, it is very confusing. The first sentence especially so. 

Thanks for the suggestion. We have revised it as best as we could

6* multiple grammar and typo errors 

Thanks for the suggestion. We will revise it as best as we can in the final draft of the manuscript.

Reviewer 2 Report

Very well designed and structured study.

Author Response

Thank you for your kind and concise review.

Reviewer 3 Report

I read with interest the article entitled "Transepithelial Enhanced Fluence Pulsed Light M Accelerated Crosslinking for Early Progressive Keratoconus with Chemically Enhanced Riboflavin Solutions and Air Room Oxygen".
I think the effort put by authors to explore new protocols for efficient trans-epithelial CXL is worth it. The reported results are interesting and I hope authors would also publish the 5 years follow un in the future.
some comments:
1. I would shortening the introduction a bit.
2. Line 136: I do not quiet get what "30 young-adult patients over 21 years " means. Can you please resentece?!
3. for All figures: I would ask to add the measuring unit for the ordinate line. Also, the figure number 4 has different color background; please adjust it accordingly 
4. In the discussion, authors could add to which patient categories they would suggest the presented protocol.

Author Response

I read with interest the article entitled "Transepithelial Enhanced Fluence Pulsed Light M Accelerated Crosslinking for Early Progressive Keratoconus with Chemically Enhanced Riboflavin Solutions and Air Room Oxygen".
I think the effort put by authors to explore new protocols for efficient trans-epithelial CXL is worth it. The reported results are interesting and I hope authors would also publish the 5 years follow un in the future.
some comments:
1. I would shortening the introduction a bit.
Thanks for the suggestion, in the new draft we will try to shorten it where it is possible.
2. Line 136: I do not quiet get what "30 young-adult patients over 21 years " means. Can you please resentece?!
Thanks for the suggestion, we modified the sentence for it for clarity.
3. for All figures: I would ask to add the measuring unit for the ordinate line. Also, the figure number 4 has different color background; please adjust it accordingly 

Thanks for the suggestion. We have changed the color of the background accordingly and added the measuring units that were missing.

4. In the discussion, authors could add to which patient categories they would suggest the presented protocol.

Thanks for the suggestion. We have adeed the patient categories suggested for this protocol in the discussion.